# Graph-Based Automatic Feature Selection for Multi-Class Classification via Mean Simplified Silhouette

## Abstract

This paper introduces a novel graph-based filter method for automatic feature selection (abbreviated as GB-AFS) for multi-class classification tasks. The method determines the minimum combination of features required to sustain prediction performance while maintaining complementary discriminating abilities between different classes. It does not require any user-defined parameters such as the number of features to select. The methodology employs the Jeffries–Matusita (JM) distance in conjunction with t-distributed Stochastic Neighbor Embedding (t-SNE) to generate a low-dimensional space reflecting how effectively each feature can differentiate between each pair of classes. The minimum number of features is selected using our newly developed Mean Simplified Silhouette (abbreviated as MSS) index, designed to evaluate the clustering results for the feature selection task. Experimental results on public data sets demonstrate the superior performance of the proposed GB-AFS over other filter-based techniques and automatic feature selection approaches. Moreover, the proposed algorithm maintained the accuracy achieved when utilizing all features, while using only 7% to 30% of the features. Consequently, this resulted in a reduction of the time needed for classifications, from 15% to 70%.

## 1 Introduction

Feature selection is a crucial step in the process of developing effective machine-learning models. Selecting the most relevant features from a data set helps to reduce model complexity, prevent overfitting, and improve model interpretability and performance (Saeys et al., 2007). In recent years, with the explosion of big data, feature selection has become an increasingly important technique in machine learning, as it can significantly reduce the time and resources required for model development and at the same time maintain prediction accuracy (Liu & Motoda, 2012).

The primary objective of feature selection is to identify the most suitable $k$-sized subset of features that can accurately depict the input data (Chandrashekar & Sahin, 2014). This process seeks to mitigate the effects of irrelevant variables and noise while preserving the accuracy of predictions (Miao & Niu, 2016). Feature selection methods are generally classified as one of three types: filter, wrapper, or embedded (Pereira et al., 2018).

*Wrapper* methods evaluate feature subsets by using a model's performance as the criterion (Li et al., 2017). These methods are computationally expensive, as they involve training and evaluating the model multiple times with different feature subsets. *Embedded* methods combine feature selection and model training into a single process. The features are selected during training, with the classifier performing the selection (Jović et al., 2015). However, the selection process depends on the specific classifier used, making it less suitable when off-the-shelf classification methods and separate feature selection algorithms are preferred. *Filter* methods rank features based on statistical properties or relevance to the target variable, but might not consider the predictive power in the context of a specific algorithm, leading to suboptimal feature selection for some models (Yang & Pedersen, 1997).

*Filter* methods apply a statistical measure to rank the features according to their relevance to the task at hand. ReliefF (Robnik-Šikonja & Kononenko, 2003), for instance, ranks the importance of each feature by measuring how well it distinguishes between instances of different classes while

considering the proximity of those instances to each other. The Fisher score method (Longford, 1987) ranks features independently based on their Fisher criterion scores, which is the ratio of inter-class separation and intra-class variance. Correlation-based feature selection (CFS) (Hall, 1999) selects features with low linear relationships with other features that are correlated with the label.

Lately, researchers have been exploring using graph-based techniques to apply filter-based feature selection methods. Graph-based techniques have emerged as a promising field due to their advantages, such as improved interpretability, capturing complex relationships between features (You et al., 2020), and the potential to handle high-dimensional data more effectively (Cai et al., 2010). These methods involve constructing a graph that captures pairwise relationships between features in the data, while also considering their relevance and redundancy. Numerous graph-based models that rank features based on specific criteria and select the $k$ features with the highest score have been proposed recently (Hashemi et al., 2020; via Invertible, 2020). A main drawback of these methods, however, is that selecting features with the highest score may result in selecting features with identical characteristics that do not cover the entire feature space, leading to a loss of information.

Friedman et al. (2023) proposed a potential solution to the aforementioned limitation through a filter-based feature selection technique that utilizes diffusion maps (Coifman & Lafon, 2006). This method encompasses the entire feature space by constructing a feature space based on the features' separation capabilities. It then selects features with complementary separation capabilities that cover the entire feature space. One major shortcoming that remains open is how to find the size of the subset of features we want to find, without relying on user input or making assumptions such as believing that a certain percentage of all features contains all necessary information or that the optimal size lies within a predetermined range.

The responsibility for determining the appropriate $k$ value for filter-based feature selection methods, however, is often delegated to the user. Thus, the latter must balance retaining valuable information and minimizing the impact of irrelevant features. This limitation poses a considerable challenge as the optimal $k$ value can vary depending on the data and the task at hand, leading to a trial-and-error approach that requires long running times and resources. Hence, there is a growing demand for a filter-based feature selection algorithm that can determine the minimal combination of features required to sustain prediction performance automatically.

Covões & Hruschka (2011), henceforth referred to as Thiago (name of the first author) technique in our paper, introduced a filter-based algorithm that employs the Simplified Silhouette (SS) index to overcome this specific limitation. The technique requires the user to specify a search range by determining both a minimum and maximum $k$, with the method subsequently identifying the optimal $k$ within this range. This approach exhibits two primary shortcomings. Firstly, if the user selects a minimum and maximum $k$ that fail to encompass the optimal $k$ value, the ideal $k$ is not identified. Secondly, although the SS index is effective for assessing clustering quality, it is not designed to evaluate clustering quality for feature selection tasks in classification problems.

In this paper, we propose a novel filter-based feature selection method for multi-class classification tasks that automatically determines the minimum combination of features required to sustain the prediction performance when using the entire feature set. We thus remove the need for user-defined parameters or for generating the classification results for every potential candidate solution. Our method defines the contribution of each feature according to its ability to separate each pair of classes, which is calculated using the Jeffries–Matusita (JM) distance. To visualize the distribution of features in relation to their ability to separate pairs of classes, we incorporate t-distributed Stochastic Neighbor Embedding (t-SNE) (Van der Maaten & Hinton, 2008), a graph-based learning technique, into our method. The method seeks to identify a set of features with complementary discrimination abilities that can separate pairs of classes, rather than selecting those with the highest separation scores as is common in other filtering methods. To do so, the K-medoids algorithm, which allows features from different areas of space to be selected, is used.

In addition, this work proposes a new index called the Mean Simplified Silhouette (MSS), which is based on the SS index (Hruschka et al., 2006). The MSS index evaluates clustering outcomes in the context of feature selection for classification problems. It assesses the effectiveness of the selected subset of features obtained from clustering results, aiming to preserve the separability between each pair of classes when using the entire feature space. The Kneedle algorithm (Satopaa et al., 2011) is then applied to the MSS values to determine the minimum set of $k$ features.

To summarize, this paper makes the following contributions:

- A graph-based feature selection method is proposed to identify the minimum set of $k$ features required to preserve the accuracy of predictions when using the entire feature set.
- A novel Silhouette-based index that can assess the diversity and complementary discrimination capabilities of the obtained features to separate each pair of classes in classification problems is offered.
- The effectiveness and superior performance of our approach compared to state-of-the-art filtering methods are demonstrated via an experimental analysis.

The remainder of this paper is organized as follows. Section 2 presents the preliminaries, Section 3 outlines our proposed model, Section 4 discusses the results and comparisons, and Section 5 concludes the paper and offers potential future directions.

## 2 PRELIMINARIES

In this section, we outline the techniques used in our proposed method. First, we present JM distance, which is used to create a new feature space based on each feature's ability to distinguish between each pair of classes. We then discuss t-SNE, a dimension-reduction technique that we use to capture complex relationships between features which are then utilized for clustering implementation. We then introduce the Silhouette index, which serves as the basis for our new MSS index aimed at assessing the quality of the clustering for the feature selection task in classification problems. Lastly, we discuss the Kneedle algorithm, which we utilize to identify the "knee" point of the MSS index curve to determine the subset of $k$ features to be used in the classification phase.

### 2.1 NOTATION

Denote the learned data set by $(X, Y)$, where $X$ is a $N \times M$ data set, with $N$ representing the number of samples and $M$ representing the features' dimension. The label is stored in the $N \times 1$ vector $Y$, which assumes $C$ classes. The data set $X$ comprises $M$ feature vectors, represented by $F = \{f_1, ..., f_M\}$, where each $f_i$ is of size $1 \times N$.

### 2.2 JEFFRIES–MATUSITA DISTANCE

The JM distance is a statistical measure of the similarity between two probability distributions (Wang et al., 2018; Tolpekin & Stein, 2009). Given a feature $f_i \in F$, we use the JM distance to construct a $C \times C$ matrix, $JM_i$, which defines how well the feature $f_i$ differentiates between all pairs of classes. Specifically, the matrix entry $JM_i(c, \tilde{c})$ indicates how well the feature $f_i$ differentiates between the two classes $c$ and $\tilde{c}$, where $1 \leq c, \tilde{c} \leq C$. The matrix entries are computed by

$$JM_i(c, \tilde{c}) = 2(1 - e^{-B_i(c,\tilde{c})}) \tag{1}$$

where

$$B_i(c, \tilde{c}) = \frac{1}{8}(\mu_{i,c} - \mu_{i,\tilde{c}})^2 \frac{2}{\sigma_{i,c}^2 + \sigma_{i,\tilde{c}}^2} + \frac{1}{2}\ln(\frac{\sigma_{i,c}^2 + \sigma_{i,\tilde{c}}^2}{2\sigma_{i,c}\sigma_{i,\tilde{c}}}) \tag{2}$$

is the Bhattacharyya distance. The values $\mu_{i,c}, \mu_{i,\tilde{c}}$ and $\sigma_{i,c}, \sigma_{i,\tilde{c}}$ are the mean and variance values of two given classes $c$ and $\tilde{c}$ from the feature $f_i$.

### 2.3 T-DISTRIBUTED STOCHASTIC NEIGHBOR EMBEDDING

t-SNE (Van der Maaten & Hinton, 2008) is a widely used nonlinear dimensionality reduction algorithm that maps high-dimensional data to a low-dimensional space while preserving local structure. The t-SNE algorithm is an improvement over the original SNE (Stochastic Neighbor Embedding) (Hinton & Roweis, 2002) algorithm, providing more accurate and interpretable visualizations by mitigating the crowding problem and simplifying the optimization process (Van Der Maaten, 2009).

The t-SNE algorithm works by embedding points from a high-dimensional space $\mathbb{R}^M$ into a lower-dimensional space $\mathbb{R}^R$, while preserving the pairwise similarities between the points ($R \ll M$).

Given a data set of $N$ points $\{u_1, u_2, ..., u_N\} \in \mathbb{R}^M$ in the high-dimensional space, the t-SNE algorithm aims to find a corresponding set of points $\{v_1, v_2, ..., v_N\} \in \mathbb{R}^R$ in the low-dimensional space that best reflects the similarities in the original space.

The algorithm defines pairwise conditional probabilities $p_{j|i}$ as the likelihood that point $u_j$ is $u_i$'s neighbor in the high-dimensional space. These probabilities are defined as:

$$p_{j|i} = \frac{\exp^{(-\|u_i - u_j\|^2/2\sigma_i^2)}}{\Sigma_{k \neq i} \exp^{(-\|u_i - u_k\|^2/2\sigma_i^2)}} \tag{3}$$

where $\sigma_i$ is the variance of the Gaussian centered at point $u_i$. The value of $p_{j|i}$ is influenced by the distance between points $u_i$ and $u_j$, with closer points having higher probabilities. The algorithm defines a symmetric pairwise similarity $p_{ij}$, which measures the similarity between points $u_i$ and $u_j$ in the high-dimensional space, defined as the average of the conditional probabilities $p_{i|j}$ and $p_{j|i}$:

$$p_{ij} = \frac{p_{i|j} + p_{j|i}}{2N} \tag{4}$$

The use of the symmetric pairwise similarity allows for a more balanced representation of similarities between points, mitigating the effects of differences in local densities. In the low-dimensional space, pairwise similarities between points $v_i$ and $v_j$ are defined as $q_{ij}$:

$$q_{ij} = \frac{(1 + \|v_i - v_j\|^2)^{-1}}{\Sigma_{k \neq l}(1 + \|v_k - v_l\|^2)^{-1}} \tag{5}$$

The t-SNE algorithm seeks to minimize the divergence between the distributions $P$ and $Q$, which is measured by the Kullback-Leibler (KL) divergence:

$$KL(P||Q) = \sum_{i \neq j} p_{ij} \log \frac{p_{ij}}{q_{ij}} \tag{6}$$

Minimizing the KL divergence ensures that the low-dimensional embedding preserves the pairwise similarities between points as accurately as possible.

## 2.4 SILHOUETTE

The Silhouette index (Rousseeuw, 1987) evaluates clustering quality by measuring how similar a data point is to its own cluster compared to other clusters. It has a value between $-1$ and $1$, indicating the level of separation between the clusters and the level of cohesion within each cluster. Specifically, the index calculates, for each point $i$, the average distance of the point from all other points in the same cluster, $a(i)$, and the average distance of the point from all other points in the closest neighboring cluster, $b(i)$. Thus, the Silhouette value for point $i$ is computed as follows:

$$sil(i) = \frac{b(i) - a(i)}{\max\{a(i), b(i)\}} \tag{7}$$

where $-1$ indicates a data point closer to the neighboring cluster, $0$ indicates a boundary point, and $1$ indicates a data point that is much closer to the other points in the same cluster than to the points of the closest cluster. The Silhouette value of a full clustering is the average value of $sil(i)$ across all data points.

The Silhouette index, being computationally expensive and sensitive to outliers, prompted the development of the Simplified Silhouette (SS) index (Hruschka et al., 2006; Wang et al., 2017), a faster and more robust alternative. The SS index for a point $i$ is computed as follows:

$$ss(i) = \frac{b(i)' - a(i)'}{\max\{a(i)', b(i)'\}} \tag{8}$$

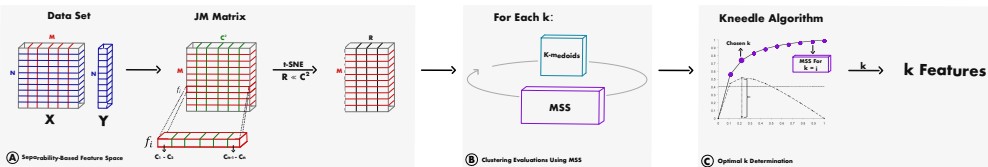

Figure 1: The GB-AFS architecture: (A) Generate a JM matrix and reduce the dimensionality. (B) Perform K-medoids clustering for every $k \in [2, M]$ and calculate MSS. (C) Find the optimal $k$ and return the corresponding $k$ features found during clustering.

where $a(i)^{'}$ is the distance of point $i$ from the centroid of its own cluster and $b(i)^{'}$ is the distance of point $i$ from the centroid of the nearest neighboring cluster (in this work, centroids replaced by medoids). The $ss(i)$ value ranges from $-1$ to $1$. Because at the end of K-means or K-medoids clustering, the distance of a data point to its closest neighboring cluster's centroid or medoid $b(i)^{'}$ is always greater than or equal to the distance to its own cluster's centroid or medoid $a(i)^{'}$, the term $\max\{a(i)^{'}, b(i)^{'}\}$ can be simplified to $b(i)^{'}$ (Wang et al., 2017). Therefore, after executing the K-means or K-medoids algorithms, the SS value for a single point can also be simplified as follows:

$$ss(i) = 1 - \frac{a(i)^{'}}{b(i)^{'}} \tag{9}$$

Similarly to the Silhouette index, the SS index is the average of the SS over all data points.

## 2.5 KNEEDLE ALGORITHM

The Kneedle algorithm (Satopaa et al., 2011) is used to identify the points of maximum curvature in a given discrete data set, commonly referred to as "knees". These knees are generally the set of points on a curve that represent local maxima if the curve is rotated by an angle of $\theta$ degrees clockwise about the point $(x_{min}, y_{min})$ through the line that connects $(x_{min}, y_{min})$ and $(x_{max}, y_{max})$ points. The identified points are those that differ most from the straight-line segment connecting the first and last data points, representing the points of maximum curvature for a discrete set of points.

## 3 METHOD

We now present our graph-based filter method for automatic feature selection (GB-AFS). As explained above, the method determines the optimal subset of features that best represent the data, achieving an effective balance between model performance and computational efficiency. The overall architecture of our method is presented in Figure 1, while Algorithm 1 outlines the specific steps for implementing the method.

### 3.1 SEPARABILITY-BASED FEATURE SPACE

Our aim is to preprocess the input data and move them into a reduced feature space that retains the original feature space's ability to distinguish between each pair of classes. For this goal, we define a new feature space by calculating the JM matrix. For each feature $i$ in $X$, we compute a $JM_i$ matrix of size $C \times C$ that captures the separation capabilities of each feature with respect to all possible pairs of classes in the input data. Each $JM_i$ matrix is reshaped to be a vector $f_i$ of size $1 \times (C^2)$ so that the JM matrix now holds the reshaped matrices as its rows. To visualize and organize the separability characteristics of the features, we use the t-SNE dimensionality reduction method. We chose t-SNE for its fast runtime, flexible perplexity parameter, and superior visualization capabilities. The compact representation of the input data in the new feature space is used as input for the clustering evaluation stage of the GB-AFS method.

### 3.2 CLUSTERING EVALUATION USING MSS

The GB-AFS aims to identify the minimal subset of $k_{min}$ features that retain the ability of the entire $M$ features to separate and distinguish different classes. To identify this subset, we follow a two-step

procedure for every $k \in [2, M]$ to obtain a score reflecting the capability of a $k$-sized feature subset to represent the entire feature space's ability to separate and distinguish between different classes.

In the first step, features are selected using the K-medoids algorithm, which selects features from different regions in the low-dimensional space, with complementary separation capabilities. The K-medoids algorithm, however, has a significant drawback in that it is sensitive to the initialization of centers. To mitigate this issue, we utilize the K-means++ initialization algorithm, which initializes the algorithm more effectively by selecting the initial centers using a probability distribution based on the distances between data points. We then move on to the next stage of the GB-AFS method. Since K-medoids is designed to minimize the sum of distances between features and the nearest medoids, the objective of the second step is to use a measure to evaluate the effectiveness of the $k$ subset of features obtained from the K-medoids algorithm in representing the entirety of the feature space by also considering the separation between clusters. Thiago (Covões & Hruschka, 2011) proposed an approach that combines K-medoids and the SS measure for the feature selection task.

We propose a new metric, named Mean Simplified Silhouette (MSS) index, which is a variation of the SS index that evaluates the clustering outcome in the context of feature selection in classification problems. Thus, we anticipate that the MSS index will have a positive correlation with classification performance. By performing multiple runs for K-medoids, we can identify the $k_{min}$ subset of features that most effectively represents the original feature set and maintain the classification performance when using the entire set of features.

---

**Algorithm 1:** Implementation of the GB-AFS method

**Input** : Feature vectors $F = \{f_1, ..., f_M\}$
**Output:** A subset of $k_{min}$ selected features

1  For each feature $f_i$ compute JM-matrix $JM_i$ of size $C \times C$
2  Reshape $JM_i$ to a $1 \times (C^2)$ vector, and construct a data set $\mathcal{Z}$ with the reshaped $JM_i$ matrices as its rows
3  Apply the t-SNE algorithm to $\mathcal{Z}$, resulting in a new data set $\mathcal{Q}$ with a lower-dimensional space $\mathbb{R}^R$
4  $S \leftarrow \emptyset$                                                    /* Store MSS values */
5  **for** $k \leftarrow 2$ *to* $M$ **do**
6  $\quad$ Apply $k$-Medoid to $\mathcal{Q}$
7  $\quad$ $S \leftarrow$ Calculate MSS value
8  $k_{min} \leftarrow$ Apply Kneedle Algorithm to $S$
9  Return the $k_{min}$ features that were found in line 6

---

### 3.2.1 MEAN SIMPLIFIED SILHOUETTE

The goal of feature selection is to identify a set of features that are represented by the accepted medoids, which can cover the entire feature space and eliminate redundant features while providing complementary capabilities. Therefore, when evaluating the output of a clustering algorithm for feature selection, the criterion should consider the distance between each omitted feature and its nearest medoid, as well as the distance between each feature and all other obtained medoids. Typically, the quality of the clustering output is evaluated using metrics such as the Silhouette index or the SS index. These metrics, however, may not be as effective when clustering is employed for feature selection in classification problems. First, they only consider the distance between every feature to the closest cluster, without taking into account the distance to all other clusters, which is very important when evaluating the complementary capabilities of the chosen features. Second, they set the value of a feature to zero (Rousseeuw, 1987; Hruschka & Covoes, 2005) if it happens to be the sole feature present in a cluster, which causes the indices to tend to zero as the number of clusters in the space approaches the number of features in the space.

Our MSS index addresses the two aforementioned drawbacks of the Silhouette and SS indices. To overcome the first issue, we consider the distances between a feature and all clusters in the feature space except the cluster to which the former belongs. This modification allows for a more reliable assessment of the selected features, ensuring that they are diverse and complementary enough to provide complete coverage of the entire feature space. To overcome the second issue, we exclude clusters that have only a single feature from the MSS calculation. The reasoning behind this is that by separating out a feature that is distant from the centroid of its cluster and creating a new cluster with just that feature should improve the clustering outcome and increase the MSS index.

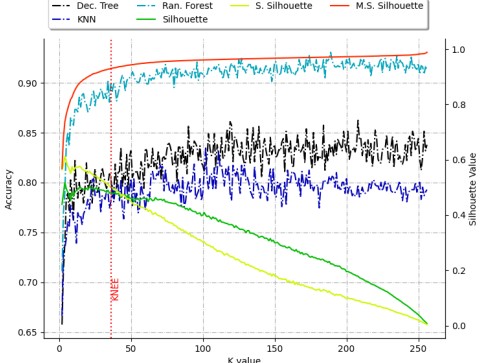

| Running Time (seconds) | | Knee Point | All Features | Time Saving |
|---|---|---|---|---|
| **Cardiotocography** (k=7) | KNN | 0.392 | 0.518 | 24.32% |
| | Decision Tree | 0.485 | 0.521 | 6.91% |
| | Random Forest | 0.523 | 0.599 | 12.68% |
| **Mice Protein Expression** (k=16) | KNN | 0.491 | 0.799 | 38.54% |
| | Decision Tree | 0.908 | 1.022 | 11.15% |
| | Random Forest | 0.909 | 1.189 | 23.54% |
| **Microsoft Malware Sample** (k=32) | KNN | 0.734 | 1.763 | 58.36% |
| | Decision Tree | 2.152 | 2.989 | 28.02% |
| | Random Forest | 2.747 | 3.723 | 26.21% |
| **Music Genre Classification** (k=33) | KNN | 0.624 | 1.383 | 54.88% |
| | Decision Tree | 2.101 | 2.624 | 19.93% |
| | Random Forest | 2.957 | 3.759 | 21.33% |
| **Isolet** (k=44) | KNN | 0.888 | 3.654 | 75.69% |
| | Decision Tree | 2.551 | 8.750 | 70.84% |
| | Random Forest | 4.415 | 11.805 | 62.60% |

Figure 2: Silhouette, SS, MSS and accuracy results obtained by three classifiers. A vertical dotted line represents the minimum $k$ value, the "knee" point found by the Kneedle algorithm.

Table 1: The running times and the percentage of time saved when using a set of $k_{min}$ features obtained by GB-AFS in comparison to the running times when using all features.

The MSS index design specifically addresses feature selection in classification problems by prioritizing a comprehensive understanding of the feature distribution throughout the entire space. It thus creates an index that is correlated with the classification results.

The MSS index is calculated based on the distances between each feature and the medoids of each cluster, similar to the SS index. It differs from the Silhouette index, which calculates distances between each feature and other features within the same cluster. We believe that our approach is preferable because it is more reasonable to calculate distances from the medoids, which are the representative features, rather than the excluded features. In addition, GB-AFS reduces the computational complexity, i.e., when computing distances from the medoids, the computational complexity is estimated as $O(kRM)$, as opposed to $O(RM^2)$, when computing distances from all features within each cluster. This difference is significant when $k$ is much smaller than the original number of features, $M$.

To compute the MSS index, we begin by defining the values $a(i)$ and $b(i)$ for every point $i$ in the data set. The value of $a(i)$ corresponds to the distance $d(\cdot, \cdot)$ between point $i$ and the center of the cluster to which it belongs, $C_h$, whereas the value of $b(i)$ denotes the average distance of point $i$ from the centers of all other clusters $C_l, l \neq h$; namely:

$$a(i) = d(x_i, C_h) \tag{10}$$

$$b(i) = \operatorname*{average}_{l \neq h} d(x_i, C_l) \tag{11}$$

$$mss(i) = 1 - \frac{a(i)}{b(i)} \tag{12}$$

where the MSS index is the average of the MSS coefficients over all data points.

Figure 2 presents the Silhouette, SS, and MSS values depicted as solid lines and the accuracy obtained by three classifiers depicted as dashed lines over different values of $k$. These results were obtained by executing the first two steps of GB-AFS (Figure 1) on the Microsoft Malware Prediction data set. It can be observed that as the value of $k$ increases, both the Silhouette and SS values decrease rapidly toward zero, indicating an increase in single-point clusters. A clear correlation was observed between the MSS index and the accuracy results of the classifiers, which emphasizes the effectiveness of the MSS index in solving feature selection tasks in classification problems.

It is important to note that our objective is not solely focused on pinpointing the $k$ value that results in the highest accuracy value. Our objective is also to find the smallest $k$ value that is sufficient for obtaining acceptable accuracy. Even if a higher $k$ value may lead to marginally better accuracy, it may demand excessive resources and computational power, which would not be practical or worthwhile. That is why we utilize the Kneedle algorithm developed by Satopaa et al. (2011) as explained in the next section. It enables us to achieve a balance between accuracy and resource usage.

Table 2: Results over the 5 data sets. A comparison of the Accuracy (left) and Balanced F-score (right) results of our method vs. state-of-the-art methods. The results of the best method are in bold for each data set and experimental setup. Paired t-test significance at $p$-value $< 0.05$ indicated by *.

| | | GB-AFS | | ReliefF | | Fisher | | CFS | | Random | |
|---|---|---|---|---|---|---|---|---|---|---|---|
| | | Accuracy | B. F-score | Accuracy | B. F-score | Accuracy | B. F-score | Accuracy | B. F-score | Accuracy | B. F-score |
| Cardiotocography (k=7) | KNN | **0.585** | 0.551 | 0.470 | 0.438 | 0.419 | 0.386 | 0.582 | **0.553** | 0.356 | 0.297 |
| | Decision Tree | **0.675 *** | **0.688 *** | 0.561 | 0.561 | 0.514 | 0.517 | 0.592 | 0.614 | 0.432 | 0.413 |
| | Random Forest | **0.746 *** | **0.761 *** | 0.633 | 0.629 | 0.582 | 0.574 | 0.679 | 0.703 | 0.525 | 0.490 |
| Mice Protein Expression (k=16) | KNN | **0.553 *** | **0.582 *** | 0.529 | 0.537 | 0.528 | 0.552 | 0.518 | 0.528 | 0.434 | 0.435 |
| | Decision Tree | 0.518 | **0.516** | **0.521** | 0.509 | 0.509 | 0.511 | 0.510 | 0.501 | 0.454 | 0.455 |
| | Random Forest | **0.696 *** | **0.690** | 0.666 | **0.690** | 0.637 | 0.650 | 0.641 | 0.643 | 0.579 | 0.472 |
| Microsoft Malware Sample (k=32) | KNN | **0.786** | 0.728 | 0.783 | **0.729** | 0.671 | 0.681 | 0.720 | 0.716 | 0.627 | 0.605 |
| | Decision Tree | **0.809 *** | **0.785 *** | 0.771 | 0.756 | 0.690 | 0.700 | 0.769 | 0.740 | 0.677 | 0.685 |
| | Random Forest | **0.900 *** | **0.795 *** | 0.833 | 0.760 | 0.743 | 0.744 | 0.763 | 0.747 | 0.664 | 0.679 |
| Music Genre Classification (k=33) | KNN | **0.476 *** | **0.551 *** | 0.414 | 0.413 | 0.437 | 0.443 | 0.391 | 0.380 | 0.280 | 0.294 |
| | Decision Tree | **0.374 *** | **0.456 *** | 0.272 | 0.290 | 0.316 | 0.331 | 0.339 | 0.381 | 0.239 | 0.248 |
| | Random Forest | **0.507 *** | **0.525 *** | 0.463 | 0.434 | 0.370 | 0.411 | 0.376 | 0.406 | 0.199 | 0.219 |
| Isolet (k=44) | KNN | **0.837** | **0.815** | 0.830 | 0.802 | 0.518 | 0.544 | 0.701 | 0.756 | 0.412 | 0.512 |
| | Decision Tree | **0.859 *** | **0.802 *** | 0.451 | 0.501 | 0.555 | 0.573 | 0.779 | 0.737 | 0.460 | 0.499 |
| | Random Forest | **0.875 *** | **0.807 *** | 0.834 | 0.779 | 0.598 | 0.685 | 0.712 | 0.711 | 0.438 | 0.503 |

## 3.3 OPTIMAL k DETERMINATION

In this third step, our goal is to determine the minimal subset of $k_{min}$ features that can classify different classes effectively without incurring a drop in performance. As presented in Section 3.2, the MSS index exhibits a correlation with the accuracy results over all possible values of $k$. Thus, in this step of the proposed GB-AFS, illustrated in Figure 1, we apply the Kneedle algorithm from Section 2.5 to the MSS graph to find the minimal subset of $k_{min}$ features.

Applying the Kneedle algorithm to the MSS graph enables the identification of the knee point, as illustrated in Figure 2 by the vertical dashed line. This knee point corresponds to a specific $k$ value representing the minimal number of features needed for classification. Subsequently, the $k$ medoids associated with this $k$ value are retrieved as the minimal subset of features required for classification.

## 4 EXPERIMENTS

### 4.1 EXPERIMENT SETUP

**Data Sets.** Our proposed method is assessed using five data sets from varied domains. You can find more details about these data sets in Appendix A.1.

**Parameter Settings.** We utilized t-SNE in our experiments following the author's recommendations (Van der Maaten & Hinton, 2008) with slight adjustments. You can find more details about these settings in Appendix A.2.

### 4.2 BASELINE METHODS

The predictive efficacy of our GB-AFS method is evaluated against a total of four filter-oriented methodologies for addressing the feature selection task: ReliefF (Robnik-Šikonja & Kononenko, 2003), Fisher Score (Longford, 1987), Correlation-based Feature Selection (CFS) (Hall, 1999), and a randomized approach where a specified number of features is chosen randomly. In each of the four benchmarked methods, we utilize the $k_{min}$ value obtained from our GB-AFS approach. We refer to these techniques as "filter methods with pre-defined $k$".

To evaluate our proposed technique for selecting the $k_{min}$ using the MSS and Kneedle algorithms, we compare it to the technique recommended in Thiago's paper, which involves selecting the $k$ value that yields the highest SS value. To accomplish this, we conducted two separate runs of the GB-AFS, each utilizing one of the two techniques, and compared the obtained results.

### 4.3 EXPERIMENT RESULTS

The details of the experimental methodology guiding this study are outlined in Appendix B. Following this, we evaluated the performance of our GB-AFS method based on two key metrics: accuracy and balanced F-score. The performance of the classifiers was determined by calculating the average values over 10 test sets. The accuracy and balanced F-score of the proposed GB-AFS compared to the filter methods with pre-defined $k$, using the same $k_{min}$ determined by GB-AFS, are reported

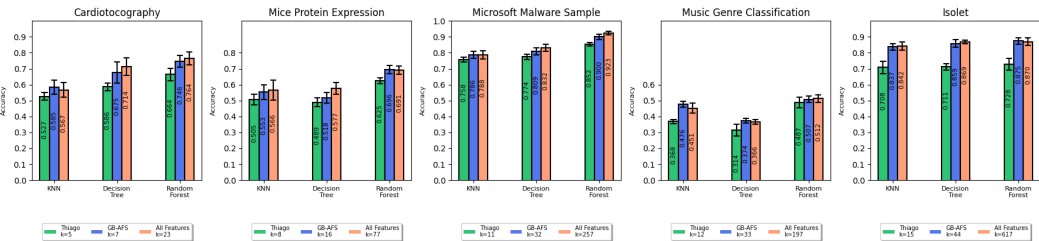

Figure 3: Comparison of the accuracy of various classifiers utilizing $k$ features selected by the GB-AFS with the MSS index, $k$ features selected by the GB-AFS using Thiago's proposed approach and the complete feature set available in the data set.

in Table 2. For each data set and classifier, the results of the best method are shown in bold. Statistically significant differences at a $p$-value $< 0.05$ based on paired t-test are indicated by *. In 14 out of 15 combinations of data sets and classifiers, we observed that the GB-AFS method performs better in terms of accuracy compared to the other methods, with improvements ranging from 0.4% to 14%, and an average improvement of 6.5%. In 11 of these 14 combinations, GB-AFS outperformed the other filter methods with a statistically significant difference ($p$-value $< 0.05$). In terms of the balanced F-score, we observe that the GB-AFS method achieved better results in 13 of the 15 combinations of data sets and classifiers, with improvements ranging from 1% to 24.4% and an average improvement of 9.5%. Of these 13 combinations, 10 were found to be statistically significant. The superior results achieved by the proposed GB-AFS method, in comparison to other baseline methods, can be attributed to its ability to identify automatically a subset of features with complementary separation capabilities. For a visual representation of this ability, please refer to the graphical analysis provided in Appendix C.

Figure 3 displays the average accuracy of the proposed GB-AFS method with a $\pm 95\%$ confidence interval, in comparison to GB-AFS method, using Thiago's approach (Covões & Hruschka, 2011) for determination of $k$ and using all features in the data set. The results show that the GB-AFS method achieved significantly better accuracy than when incorporating Thiago's method for determination of $k$, with improvements ranging from 3.7% to 29.3%, and an average improvement of 12.7%. Moreover, the GB-AFS method selected between 7% and 30% of the features in each data set. In 14 of 15 combinations of data sets and classifiers, there was no statistically significant difference ($p$-value $< 0.05$) in accuracy results between the GB-AFS method and when using all features. While the accuracy results are similar, the percentage of time saved on average by using a set of $k_{min}$ features ranges from 15% for the smallest data set to 70% for the largest (Table 1).

## 5 CONCLUSIONS AND FUTURE WORK

This paper presents a novel graph-based filter method for automatic feature selection (GB-AFS) for multi-class classification problems. An algorithm is developed to apply the proposed method to find the minimal subset of features that are required to retain the ability to distinguish between each pair of classes. The experimental results on five data sets show that the proposed algorithm outperformed other filter methods with an average accuracy improvement of 6.5%. Moreover, in 14 of 15 cases, the GB-AFS method was able to identify 7% to 30% of the features that retained the same level of accuracy as when using all features, while reducing the classification time by 15% to 70%.

Future work could utilize the proposed method for constraint-based classification problems. In many real-world situations, each feature incurs an economic cost for collection, with limited resources being available to tackle the problem at hand. Thus, the adaption of the proposed algorithm, such that the total cost of the selected features meets the constraint, is an interesting direction to explore. Additionally, it would be worthwhile investigating the potential of incorporating asymmetric error costs between pairs of classes into the proposed method.

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

## A  ADDITIONAL EXPERIMENT SETUP DETAILS

### A.1  DATA SETS

We use five data sets from different domains. Table 3 presents, for each data set, the number of instances, the number of features, and the number of classes. Below is a short description of each data set.

***Isolet*** is a data set of 617 voice recording features from 150 subjects reciting the English alphabet, with the goal of classifying the correct letter among the 26 classes.

***Cardiotocography*** comprises 23 distinct assessments of fetal heart rate (FHR) and uterine contraction (UC) characteristics, as documented on cardiotocograms. They were categorized into 10 separate classes by experienced obstetricians.

***Mice Protein Expression*** measures the expression levels of 77 proteins in the cerebral cortex of eight classes of mice undergoing context fear conditioning for evaluating associative learning.

Table 3: Overview of the data sets

| Data Set | #Instances | #Classes | #Features |
|---|---|---|---|
| Isolet (Dua & Graff, 2017) | 7797 | 26 | 617 |
| Cardiotocography (Dua & Graff, 2017) | 2126 | 10 | 23 |
| Music Genre Classification (Olteanu, 2020) | 1000 | 10 | 197 |
| Microsoft Malware Sample (Microsoft, 2019) | 1642 | 9 | 257 |
| Mice Protein Expression (Higuera et al., 2015) | 1080 | 8 | 77 |

***Microsoft Malware Prediction***, released in 2018, is a comprehensive collection containing 257 file attributes and nine malware identification classes. It was created by Microsoft to facilitate the development of machine-learning models for predicting malware.

***Music Genre Classification*** is a data set of 1,000 labeled audio segments, each lasting 30 seconds and containing 197 features. Covering 10 distinct genres, this data set is frequently employed in machine-learning projects aimed at classifying music genres.

## A.2    PARAMETER SETTINGS

We utilized t-SNE in our experiments according to the author's Van der Maaten & Hinton (2008) recommendations, which suggest setting the number of iterations to 1000 and using a perplexity range between 5 and 50. We employed two-dimensional parameter settings exclusively for visualization purposes. For each data set, we chose a perplexity value of 30, except for the Cardiotocography data set, where a value of 10 was applied.

For K-medoids clustering, we used the PAM method for cluster assignment with K-means++ initialization. This method selects initial medoids farthest from each other, simulating the idea of choosing features with complementary separation capabilities, which improves the algorithm's efficiency and accuracy.

## B    EXPERIMENT METHODOLOGY

To avoid attributes with large numerical ranges from dominating those with small numerical ranges, the data were rescaled to lie between 0 and 1 using the min–max normalization procedure. Then, we split the data randomly such that $75\%$ of the instances (training data set) were used for applying the GB-AFS to determine the set of $k_{min}$ features and build the classifiers. The remaining $25\%$ (test data set) were used to evaluate the performance of the GB-AFS and resulting classifiers.

To find the $k_{min}$, we evaluated the MSS over a validation data set for each set of $k$ features found by applying the GB-AFS on the training set, where $k \in [2, M]$. To reduce the bias when selecting the training and validation data, we used a five-fold cross-validation approach (Arlot & Celisse, 2010), where 80% of the data set was used to identify the set of $k$ features and the remaining 20% was used for MSS calculation. For each value of $k$, we calculated the MSS five times, each time using a different subset as the validation data set. We then averaged these values to generate an averaged MSS graph. Next, we applied the Kneedle algorithm[1] to the averaged MSS graph to obtain the value of $k_{min}$, which represents the number of features in the final data set.

After determining $k_{min}$, we applied the GB-AFS to the entire training set to obtain the set of $k_{min}$ features and construct three different classifiers (KNN, Decision Tree and Random Forest) based on the chosen features. It should be noted that these classifiers were chosen to enable the evaluation of our method in relation to other methods. They are not part of the method we propose, as can be seen in Figure 1; other classifiers could just have easily been used. The trained classifiers were then employed to classify instances in the out-of-sample test set and evaluate the accuracy and balanced F-score metrics. To evaluate the statistical significance of the results in comparison to the benchmarked methods, we repeated the entire experiment's methodology 10 times, using a different random split of the training–test sets for each iteration.

---

[1]We use this repository that implement the Kneedle algorithm: `https://github.com/arvkevi/kneed`

## C GRAPHICAL ANALYSIS AND INTERPRETATION

Figure 4 shows the features of the Microsoft Malware data set embedded in low-dimensional space accepted by the t-SNE method. Each feature $i$ is assigned a color based on the mean value of its computed $JM_i$ matrix, in such a way that the higher the value, the darker the color tends to be. The $k_{min}$ subset of features chosen by the ReliefF method and the GB-AFS method, assuming that $k_{min}$ was found by the proposed method, are indicated by a rhombus and a circle, respectively. The ReliefF method tends to select features with a high JM mean value, whereas the GB-AFS method selects features that complement each other in terms of class separability. A similar behavior was observed in Friedman et al. (2023) when utilizing diffusion maps together with K-means.

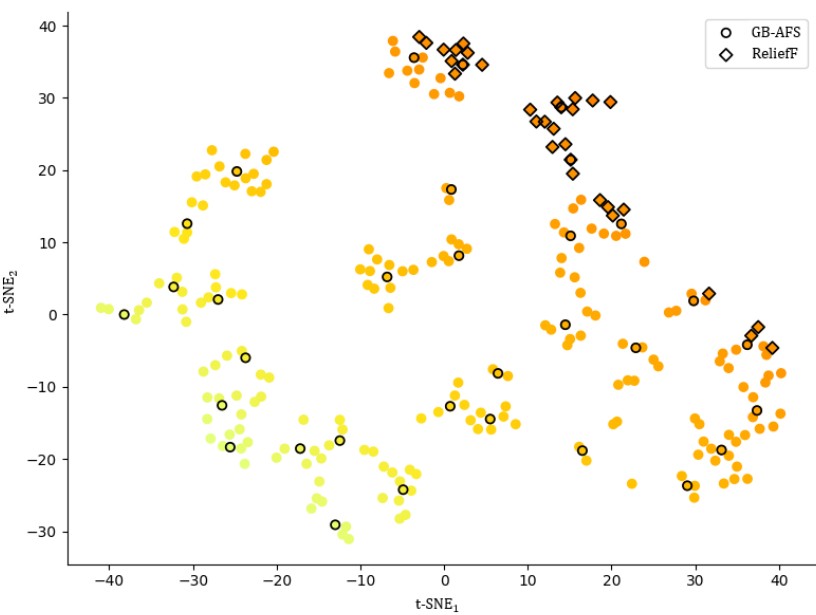

Figure 4: Microsoft Malware Sample features are color-coded according to their average JM score in t-SNE. Our algorithm found $\frac{32}{257}$ features based on their complementary separation abilities, while ReliefF marked features with high JM mean values and not necessarily with complementary abilities.

The clearly defined minimized feature landscape stands as a witness to the organized pattern found within the lower-dimensional feature arena. To exemplify this, we will showcase it utilizing the Microsoft Malware Sample data set, depicted in Figure 5. Through the adept organization within this minimized space, one can note the prominent separation properties, affirming the embedding's quality, which retains the local distances evident in the original JM matrices.

This data set is characterized by a composition of 257 features segmented into 9 classes, hence forming a representation through 257 JM matrices, each manifesting as a 9×9 grid. Each grid element articulates the separation degree between two distinct classes, with a pronounced separation depicted in red and gradually transitioning to yellow as the separation narrows. A feature's hue is determined by the mean value encapsulated within its respective JM matrix.

Upon closer examination, it becomes apparent that features bearing similar JM matrices tend to cluster in proximity within this diminished feature space, elucidating the effectiveness of employing t-SNE techniques on JM matrices. This approach excels in extracting separation details, facilitating a thoughtful arrangement of features in alignment with their separation attributes – a foundational principle of the introduced GB-AFS methodology.

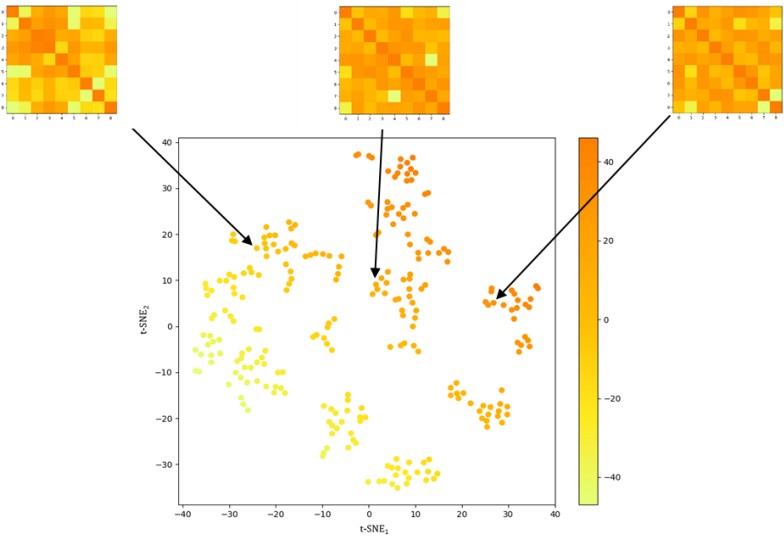

Figure 5: Application of the Separability-Based Feature Space part of the GB-AFS method to 257 features from the Microsoft Malware Sample dataset. The features are embedded to a 2-dimensional space by t-SNE and are colored by their average JM value.

