# OpenReview forum: "Graph-Based Automatic Feature Selection for Multi-Class Classification via Mean Simplified Silhouette"
_ICLR.cc/2024/Conference — ICLR 2024 Conference Withdrawn Submission_

### Official Review · Reviewer_uTMn · 2023-10-16

**Soundness:** 2 fair
**Presentation:** 2 fair
**Contribution:** 2 fair
**Rating:** 3
**Confidence:** 4

**Summary:**

In this paper, authors propose a novel filter-based feature selection method for multi-class classification tasks that automatically determines the minimum combination of features required to sustain the prediction performance when using the entire feature set.

**Strengths:**

1. The problem focused in this paper, i.e., feature selection for multi-class classification, is meaningful.

2. A novel filter-based feature selection method is proposed for multi-class classification tasks, which can automatically determine the minimum combination of features required to sustain the prediction performance when using the entire feature set.

3. Experiments validate the effectiveness of the proposed method.

**Weaknesses:**

The main problem of this submission is its novelty.

On one hand, as stated in Introduction, "A graph-based feature selection method is proposed to identify the minimum set of k features required to preserve the accuracy of predictions when using the entire feature set". However, in my opinion, the graph-based technique is due to the tSNE algorithm which is used as a part of the proposed method, i.e., the step 3 in Algorithm 1.

On the other hand,  as stated in Introduction, "A main drawback of these methods, however, is that selecting features with the highest score may result in selecting features with identical characteristics that do not cover the entire feature space, leading to a loss of information" and "In this paper, we propose a novel filter-based feature selection method for multi-class classification tasks that automatically determines the minimum combination of features required to sustain the prediction performance when using the entire feature set". I do not think there are some connections between these two statements. Moreover, how to verify the obtained $k_{min}$ in step 8 of Algorithm 1 is the minimum combination of features?

In experiments, more state-of-the-art methods should be used as baselines. All the three methods, i.e., ReliefF, Fisher Score, Correlation-based Feature Selection (CFS), are more than 20 years old. Such comparison cannot reflect the effectiveness of the propose approach.

**Questions:**

If authors disagree my comments in weaknesses, clarifications can be presented in the rebuttal phase.

---

### Official Review · Reviewer_8xaB · 2023-10-31

**Soundness:** 2 fair
**Presentation:** 3 good
**Contribution:** 1 poor
**Rating:** 3
**Confidence:** 5

**Summary:**

This article proposes an approach for a filter based feature selection, with the goal of selecting small(er) number of features that are sufficient for a building a high-performing multi-class classification model. The procedure is to use Jeffries-Mautista Distance to construct the  matrix encapsulating the discriminativeness of (each) feature for (every) pair of classes. T-SNE is then used to reduce dimensionality of this expanded feature representation, and k-medoids clustering is then applied to group the features. A newly proposed Mean Simplified Silhouette metric is used to measure the quality of clustering, and Kneedle algorithm is then used to find the best k for clustering based on the shape of the graph with Mean Simplified Silhouette Metric for different k's. After selecting the optimal k and corresponding features, three common classification algorithms: KNN, Decision Trees and Random Forests; were constructed on 5 datasets and performances were compared against four alternative approaches: Relief, Fisher Score, Correlation Based Feature Selection and Random feature selection. Proposed Graph-Based Automatic Feature Selection showed increased predictive performance in terms of accuracy and balanced F-score.

**Strengths:**

Paper is written clearly and flow is smooth. The biggest novelty is the introduction of Mean Simplified Silhouette index, which compared to the original Silhouette Index is taking into account the datapoints in all the other clusters, not just in the nearest competitor one. Also, it avoids the singleton clusters having the value of zero, which would happen for the higher values of the K. Predictive performance on the five benchmark datasets is better than the alternative approaches, and computation time savings of using reduced number of features is quantified.

**Weaknesses:**

One weakness is apparent lack of novelty. Jeffries-Mautista Distance has allready been applied in the feature selection algorithms. Kneedle Algorithm can also be adopted on arbitrary filter based feature selection method. The Mean Simplified Silhouette is novel but fairly incremental change.
Another weakness is in the empirical evaluation of the approach. The 'state of the art' competitor filter methods are very classic/basic ones, for example no more recent approach that uses Intercooperation measures is included (like Zhao, Zheng, and Huan Liu. "Searching for interacting features in subset selection." , Singha, Sumanta, and Prakash P. Shenoy. "An adaptive heuristic for feature selection based on complementarity." etc.).
The necessity and effects of dimensionality reduction with T-SNE are not clear/evaluated. In fact there is no Ablation study to determine the utility of each of the steps, and especially because at each step a number of candidates can be used: eg Mutual Information instead JMD; PCA instead of t-SNE, etc.

**Questions:**

I suggest addition of more recent filter approaches, using of some classifiers of artificial neural network type, and ablation study to see how each of the proposed steps are contributing as well if the chosen component is the best among the competitors.

---

### Official Review · Reviewer_yuKD · 2023-10-31

**Soundness:** 2 fair
**Presentation:** 2 fair
**Contribution:** 3 good
**Rating:** 6
**Confidence:** 3

**Summary:**

The paper proposed a new graph-based filter method for automatic feature selection (GB-AFS) for multi-class classification. This method combines the Jeffries-Matusita (JM) distance and t-distributed Stochastic Neighbor Embedding (t-SNE) to project a space that could capture the differentiation between the classes.

**Strengths:**

The paper introduces the GB-AFS method, a novel graph-based feature selection method, and presents the unique JM matrix and MSS index for improved separability and evaluation.
Structured methodology, validated through experimental results the compared methods.
Well-written with clear explanations of new concepts.

**Weaknesses:**

The introduction of the JM matrix and MSS index is intriguing, but considering their theoretical foundation and why they specifically lead to improved separability is not deeply explored. More detailed explanations or clearer justifications would highlight their importance. The paper lacks a time comparison of feature selection algorithms.
It's unclear how the GB-AFS method scales with larger datasets. Performance metrics and computational times for larger datasets would provide clarity on its scalability.

**Questions:**

* Could the authors provide more insight into the use of the JM matrix? Specifically, what are the advantages of using that in graph-based feature selection?

* The paper seems to miss a time comparison of the feature selection algorithms. How does the GB-AFS method fare in terms of computational time compared to other filter-based algorithms?

* Regarding Figure 4 in the appendix, could you elaborate on what each point represents?
In the statement "Each feature i is assigned a color based on the mean value of its computed JMi matrix, in such a way that the higher the value, the darker the color tends to be."
Shouldn't each point represent a sample rather than a feature in Figure 4?

---

### Official Review · Reviewer_sV3v · 2023-11-01

**Soundness:** 3 good
**Presentation:** 4 excellent
**Contribution:** 3 good
**Rating:** 5
**Confidence:** 5

**Summary:**

The paper proposed a new method GB-AFS for automatic feature selection in multi-label classification. It leverages the JM distance per class as coordinate and maps each feature to its ‘feature space’. Then it employs t-SNE to project to a lower dimensional space for clustering. It also proposes a way to pick the optimal k utilizing MSS and kneedle algorithm. The authors also conducted experiments to visualize the comparison between GB-AFS and other approaches and yield impressive result.

**Strengths:**

1. The paper is well written.
(placeholder for future edit, please allow me an extra day to finish the writing)

**Weaknesses:**

(placeholder for future edit, please allow me an extra day to finish the writing)

**Questions:**

1. P9 Figure 3. Are the metrics (accuracy and f score) computed via a set of feature selection methods using the same k picked by MSS or it just GB-AFS running over several test sets?

(placeholder for future edit, please allow me an extra day to finish the writing)